# Does Previous Pelvic Organ Prolapse Surgery Influence the Effectiveness of the Sub-Urethral Sling Procedure

**DOI:** 10.3390/jcm9030653

**Published:** 2020-02-28

**Authors:** Edyta Horosz, Andrzej Pomian, Aneta Zwierzchowska, Wojciech Majkusiak, Paweł Tomasik, Ewa Barcz

**Affiliations:** Department of Obstetrics and Gynecology, Multidisciplinary Hospital Warsaw-Miedzylesie, 04-749 Warsaw, Poland; edytahorosz@tlen.pl (E.H.); apomian@gmail.com (A.P.); a.j.zwierzchowska@gmail.com (A.Z.); wmajkusiak@gmail.com (W.M.); p_tomasik@wp.pl (P.T.)

**Keywords:** treatment, quality of life, pelvic organ prolapse, urinary incontinence

## Abstract

Pelvic organ prolapse (POP) often co-occurs with stress urinary incontinence. There is no consensus on whether prolapse repair and anti-incontinence surgery should be performed concomitantly or separately, in a two-step manner. The present study evaluated the effects of the tension-free vaginal tape (TVT) procedure in patients who had previously undergone pelvic floor repair (study group), compared to women who underwent TVT insertion only (control group). The study group comprised 84 patients who underwent the TVT procedure but had previously also undergone surgical POP repair. The control group consisted of 250 women in whom the TVT was inserted. The primary objective was to compare the objective cure rate and the secondary objective was to compare the subjective cure rate in both groups. Negative pad test was achieved in over 91% in both groups. Objective and subjective cure rates were compared, as well as complication rates. Significant improvement was observed in the postoperative 1-h pad test in all patients. In all patients, we observed significant improvement in the quality of life, with no differences between the groups. No differences were found in the occurrence of postoperative urinary retention, urgency and frequency of daytime micturition, or vaginal erosion between the groups. The current results demonstrate that the two-step approach to pelvic reconstruction and anti-incontinence surgery is as safe and effective as primary TVT implantation.

## 1. Introduction

Stress urinary incontinence (SUI) is one of the most common health problems in the adult female population. It affects quality of life in many aspects and influences socioeconomical status [1]. Since surgery is considered the most effective treatment option, there has been an ongoing discussion regarding the results of the procedures in different treatment modalities and in various medical circumstances, as well as in women with specific comorbidities.

Urinary incontinence is often accompanied by other pelvic floor disorders, mainly pelvic organ prolapse [2]. The co-existence of those pathologies is associated with the fact that there are certain pathogenetic risk factors that these two entities have in common, i.e., delivery trauma of pelvic floor muscles and fascias. Approximately 40% of woman who undergo prolapse surgery report urinary incontinence postoperatively [3].

Prophylaxis is obviously the best way to avoid pelvic floor disorders, but it is only possible during hysterectomy [4]. De novo SUI occurs in ca. 20% of patients, but among women in whom leakage is observed during the preoperative stress test with the prolapse repositioned (occult stress incontinence, OSUI), the risk of SUI may be greater than 60% and is connected with anatomical correction of urethral kinking or compression caused by advanced prolapse [3]. It should be stressed that in some cases we cannot predict UI after POP repair.

Until now, no consensus has been established regarding the question whether pelvic reconstruction procedures and anti-incontinence surgery should be performed concomitantly or not and what the order of the procedures should be. The main rationale for concomitant procedures is that this approach may reduce incontinence rates after POP repair in both symptomatic and asymptomatic women [5,6]. It has been demonstrated that it may help reduce both subjective and objective symptoms of incontinence and LUTS (lower urinary tract symptoms) [7]. On the other hand, combining POP repair with anti-incontinence surgery may lead to overtreatment and cause serious adverse events (SAE), i.e., bleeding, bladder perforation, urinary tract infection, long-term pain and erosion, or incomplete bladder voiding [8,9].

In the current literature, data concerning the effectiveness of sling procedures performed in patients who had previously undergone POP repair are lacking. Evidence supporting the value of combination surgery is also limited. Therefore, it is difficult to balance the risks and benefits of prolapse repair with the addition of a mid-urethral sling (MUS). Having acknowledged that combined surgery may be associated with higher rates of complications requiring invasive procedures or reoperation, there still exists the question of whether it is more reasonable to perform the POP and anti-incontinence procedure in a two-step manner.

The abovementioned considerations are of great importance since we should provide patients with adequate information concerning cure rates in cases where there is a necessity of POP repair and an anti-incontinence procedure.

Taking into consideration the higher percentage of complications and lower cure rates of simultaneously performed procedures, the aim of the present study was to evaluate the subjective and objective cure rates as well as the complication rates of the retropubic sling procedure in patients who had previously undergone pelvic floor reconstruction surgery (a two-step procedure) compared with patients who underwent the anti-incontinence sling procedure only. The goal of the current study was also to evaluate the possible advantages of this two-step approach compared to the one-step surgery described in the literature.

## 2. Materials and Methods

The study was performed between 2017 and 2019 as a prospective cohort study. The study group comprised 84 consecutive patients who underwent the TVT sling procedure but had also previously had a surgical POP repair in our department. Fifty-six of those women had presented SUI symptoms before POP correction. The control group consisted of 250 consecutive cases in whom the TVT (tension-free vaginal tape) was inserted. There were no differences in POPQ between the groups (mean 1.4+/0.5) before the anti-incontinence procedure.

Patients in the study group were statistically significantly older than the women in the control group. They had also had a greater average number of vaginal deliveries. No other demographic differences between the groups were detected (Table 1).

The time of observation after sling insertion was 6–12 months. All patients who underwent the TVT (primary or after POP reconstruction) procedure had no pelvic organ prolapse over POPQ 2 in any compartment. Patients who had previously undergone POP repair were then divided into subgroups, based on the types of reconstruction (Table 2). Before the reconstruction procedure we had used the Pelvic Organ Prolapse Quantification System (POPQ) to assess the prolapse degree. In patients with a symptomatic isolated defect of the anterior or posterior compartment stage ≥ 2, native tissue reconstructive surgery had been performed. In women with uterovaginal prolapse with apical compartment stage ≥ 3, laparoscopic sacrocolpopexy or six-point fixation vaginal mesh (InGynious, A.M.I., Feldkirch, Austria) had been performed. There was also a group of patients who had undergone more than one POP repair procedure (Table 2). All patients who underwent SUI surgery had the retropubic sling implanted (Gynecare TVT blue, Ethicon Inc., Johnson & Johnson, Somerville, NJ, USA). In all cases SUI was confirmed with medical history and cough stress test in semi-sitting position with bladder filling ca. 300 mL (assessed with ultrasound). Pre- and postoperative examination included a 1-h pad test as well as pelvic examination with a POPQ scale assessment. Patients in whom pelvic floor reconstruction had been performed were examined a minimum of three months after prolapse repair and then a decision considering the anti-incontinence surgery was made. Surgical procedure of TVT implantation was performed according to the 1/3 rule, after pelvic floor ultrasound examination of the urethral length. Tensioning of the tape was achieved with intraoperative cough test [10].

All postmenopausal patients (87%) received vaginal estrogen for a minimum of six weeks before and up to three months after the surgery.

The quality of life was assessed pre- and postoperatively with the use of the IIQ-7 test (Incontinence Impact Questionnaire 7). IIQ-7 is a seven-item self-report that measures disease-specific quality of life in women suffering from SUI. It contains questions concerning the impact of SUI on the ability to do household chores, engage in physical recreation and entertainment activities, travel by car or bus more than 30 min from home, participate in social activities, as well as on emotional health (e.g., feeling frustrated). In all the patients PVR (post-void residual) was checked before and the day after surgery. Postoperative urinary retention, overactive bladder (OAB) symptoms, possible pain and infection symptoms were assessed in all patients 6–12 months after the sling procedure. 

The primary outcome of the study was the objective cure, defined as no leakage during a cough test with bladder filling ca. 300 mL and negative 1-h pad test (≤2 g). Objective cure rates (1-h pad test results) were compared between the control group and the study group, with the consideration of the type of previous POP surgery. The secondary outcome was the subjective cure assessed with the use of quality of life questionnaires. We also analyzed intra- and post-operative complications, i.e., bladder perforation, urinary retention (post-voiding residual volume > 100 mL), vaginal mucosal erosion, and de novo OAB. 

The local ethics committee approved the study.

Descriptive statistical analysis and statistical tests were performed using R version 3.4.0 (by the R Foundation for Statistical Computing, Vienna, Austria). A U Mann–Whitney test was used to compare quantitative variables. The Kruskal–Wallis test was used to test for differences within three or more subgroups of a variable. A *p*-value < 0.05 was considered significant. For categorical data, chi-square and Fisher’s exact tests were used.

## 3. Results

A total of 334 patients were enrolled in the study, 84 patients who had previously undergone POP repair (study group) and 250 who underwent primary TVT (control group).

Women from the study group had undergone the following types of POP reconstructive surgery: sacrocolpopexy, native tissue repair, mesh implantation, or a combination of different types of surgeries (Table 2).

We evaluated the pre-operative 1-h pad test results in the study and control groups. No statistically significant differences were observed between the groups. Similarly, no differences were reported in the Incontinence Impact Questionnaire 7 scores (Table 3 and Table 4).

The main goal of our study was to evaluate objective cure rates in both groups and to show whether there were differences between patients who underwent POP repair before the TVT procedure and patients who underwent the primary sling procedure. Statistically significant improvement was observed in the postoperative 1-h pad test in all patients; this parameter did not differ between the groups. In both groups, a negative pad test was achieved in over 91% of patients (Table 5). In all patients, we observed a significant improvement in the quality of life, with no differences between the study and control group (Table 6). The secondary objective of the study was to compare the adverse effects of both schemes, primarily OAB de novo incidence. Therefore, we compared the symptoms of OAB in both groups. No differences were found in the postoperative urinary retention or urgency and frequency of daytime micturition between the groups. We observed more episodes of nighttime micturition in the study group (Table 7). In none of the patients from either group were major intraoperative bleeding, bladder perforation, or vaginal mucosal erosions reported.

## 4. Discussion

Surgical treatment is the preferred option for women with stress urinary incontinence in whom conservative management strategies have failed [11]. According to the available literature, mid-urethral slings are both more effective and safer than other surgical treatment modalities [12], with objective and subjective cure rates of retropubic TVT ranging from 71% to 97% [13,14]. There has also been an ongoing discussion considering other treatment options such as fascial slings.

In the present study, we aimed to evaluate the effectiveness and cure rates of the retropubic MUS in patients who had previously undergone prolapse surgical repair compared to patients who underwent anti-incontinence surgery only in order to evaluate the rationale for two-step procedure in patients with POP and stress urinary incontinence.

As there is a high co-occurrence of pelvic organ prolapse and urinary incontinence, some surgeons decide to perform concomitant procedures, with good results. A retrospective cohort analysis of 102 patients who underwent the TVT obturator system implantation in conjunction with a variety of POP repair surgeries revealed that, regardless of the POP operation type, patients demonstrated improvement in validated SUI outcomes during the following two years [7]. Chai et al. demonstrated that concomitant surgery at the time of mid-urethral sling implantation (MUS) did not increase complication rates, and women receiving such model of treatment had a significantly lower risk of objective failure after MUS [15].

On the other hand, prophylactic anti-incontinence procedures performed at the time of prolapse repair lead to apparent overtreatment. A randomized controlled study of prolapse surgery with or without a TVT in women with preoperative occult stress incontinence or asymptomatic urodynamic incontinence showed that only 7% of patients in the group that had undergone only POP repair requested sling insertion six months after the prolapse repair. The authors estimated that six to 11 slings would have to be inserted to prevent one woman from becoming stress-incontinent after prolapse surgery [16]. Similar results were obtained by Wei et al., who demonstrated that combined prolapse and anti-incontinence surgery in women without preoperative symptoms of SUI reduces the likelihood of SUI three and 12 months after surgery, but the number of patients who needed to be treated with a sling to prevent one case of urinary incontinence or treatment of incontinence at three months was 3.9, whereas at 12 months it was 6.3 [9].

Women with POP and coexisting or occult SUI have the highest risk of postoperative SUI [7,16]; however, the predictive value of demonstrable SUI is limited and varies between studies [17,18]. According to some authors, the use of a vaginal pessary can help detect occult SUI and decide whether the insertion of a MUS should be done on top of the prolapse surgery [19], but in the opinion of others occult SUI detection rates depend on the technique applied and can significantly vary, from 6% with pessary prolapse reduction, to 16% with manual, to 30% when using a speculum [17].

Moreover, positive preoperative stress test is not highly predictive of the occurrence of postoperative SUI, with no benefit of routinely added urodynamics, and it is not sufficient to determine the need for anti-incontinence surgery at the time of prolapse repair [20,21,22].

It also must be considered that concomitant prolapse and anti-incontinence surgery can cause more complications and adverse effects than two-step procedures. Wei et al. demonstrated higher rates of bladder perforation, urinary tract infection, major bleeding complications, and incomplete bladder voiding in patients with retropubic MUS implanted at the time of the prolapse repair compared to the group that underwent POP surgery only [9].

The Matsuoka’s metaanalysis demonstrated that there were higher rates of intraoperative complications in the case of combined surgery, primarily associated with bladder perforation, major bleeding, and even bowel or obturator nerve injury [23]. Boysen et al. analyzed 4793 patients who underwent abdominal sacrocolpopexy with concurrent sling placement performed in 1627 women and found that combined surgery was associated with higher rates of postoperative urinary tract infection [24].

The literature review showed a lower risk of having postoperative SUI in women with preoperative SUI symptoms and occult SUI who underwent vaginal prolapse repair with concomitant MUS implantation, but serious adverse events (SAE—events requiring an invasive procedure or reoperation or resulting in the failure of one or more organ systems or death) were more frequent in the MUS group (14% versus 8%). SAEs included bladder perforation, urethral injuries, tape exposures, MUS-related pain and erosion, and long-term voiding difficulties resulting in tape removal. Studies that analyzed general health and quality of life showed no differences between vaginal prolapse repair with or without MUS [21].

Taking the abovementioned data into consideration, we face a very important clinical question: is it reasonable to perform concomitant POP and anti-incontinence surgeries in all patients? If we decided to do so, we would have to accept the increased risk of complications, both in women who would and those who would not require the sling procedure. In the current literature, there is a lack of evidence that the two-step procedure is as safe and effective as a primary sling procedure and therefore should be recommended instead of concomitant POP repair and anti-incontinence surgery.

In the present study, we obtained the same high objective cure rate in patients who had previously undergone POP repair as in those who underwent primary sling implantation. The results were not influenced by the type of prolapse reconstruction.

Contrary to the cited authors, we did not observe higher rates of complications such as excessive bleeding or bladder perforation (0% in both groups). We also did not report significant differences in overactive bladder symptoms such as pollakisuria, nocturia, or urgency. OAB de novo after the POP repair and anti-incontinence procedure performed simultaneously might be connected with a restriction of bladder neck mobility, as was observed by Huang et al. [25]. In the case of the two-step procedure, bladder neck mobility may be assessed before the sling procedure in the stable anatomical situation to avoid such a complication.

The current results allow us to postulate that the two-step procedure is safer than the combined surgeries as far as intraoperative complications, urinary retention, and de novo OAB are concerned. An important rationale for avoiding concomitant surgeries is the observation of Ugianskiene et al. that almost 50% of patients no longer have urine leakage after only POP repair [26].

The strengths of our study include a homogenous population and standardized surgical techniques. Moreover, the data concerning the results of combined surgical treatment compared to a two-step procedure are scant. We have opted for the two-step approach since we aim to avoid unnecessary complications and overtreatment. This issue is particularly significant nowadays, when many doubts concerning the use of synthetic materials and the sling procedure in urogynecology have emerged in both the professional literature and the mass media.

## 5. Conclusions

The current results demonstrate that the two-step approach to pelvic reconstruction and an anti-incontinence surgery (retropubic sling) is as safe and effective as primary MUS implantation, regardless of the type of previous POP repair. We suggest that clinicians should opt for this method in order to avoid overtreatment and complications, especially in patients in whom SUI would not occur after the POP repair.

## Figures and Tables

**Table 1 jcm-09-00653-t001:** Demographic characteristics.

	Control Group	Study Group	*p*
Age	57.1 ± 10.9	60.0 ± 12.0	<0.001
Height	162.9 ± 6.2	163.6 ± 6.3	Ns
Body weight	72.7 kg ± 11.8	74.7 ± 11.6	Ns
Body mass index (BMI)	27.4 ± 4.3	27.9 ± 4.1	Ns
No. of deliveries (total)	2.0 ± 0.8	2.3 ± 0.9	Ns
No. of vaginal deliveries	1.8 ± 0.9	2.2 ± 0.9	<0.001
No. of C-sections	0.2 ± 0.5	0.1 ± 0.3	Ns
No. of instrumental deliveries (vacuum or forceps)	0.0 ± 0.1	0.0 ± 0.0	Ns
Average birth weight	3360 g ± 490	3443.7 ± 454.5	Ns
Maximum birth weight	3575 g ± 523	3650 ± 497.6	Ns

Data are given as mean, ± SD.

**Table 2 jcm-09-00653-t002:** Types of POP (Pelvic organ prolapse.) repair in the study group.

Type of Surgery	No. of Patients
Sacrocolpopexy	12
Native tissue reconstructive surgery	33
Vaginal mesh	24
Sacrocolpopexy + native tissue surgery	5
Mesh + native tissue surgery	8

**Table 3 jcm-09-00653-t003:** The results of the 1-h pad test in both groups before sling implantation.

	Control Group	Study Group	*p*	Sacrocolpopexy	Native Tissue Repair	Vaginal Mesh	Sacrocolpopexy with Native Tissue Repair	Vaginal Mesh and Native Tissue Repair	*p*
Pad test (g)	71.3 ± 71.6	88.9 ± 81.7	0.052	119.9 ± 113.7	85.5 ± 74	71.6 ± 66.8	92.8 ± 111.8	101.8 ± 84.8	0.74

Data are given as mean, ± SD.

**Table 4 jcm-09-00653-t004:** The results of the IIQ-7 questionnaire in both groups before sling implantation.

	Control Group	Study Group	*p*	Sacrocolpopexy	Native Tissue Repair	Vaginal Mesh	Sacrocolpopexy with Native Tissue Repair	Vaginal Mesh and Native Tissue Repair	*p*
IIQ-7 score	74.3 ± 16.2	71.9 ± 24.3	0.81	82.4 ± 16.7	77.1 ± 15.2	60.5 ± 30.5	59.0 ± 43.3	83.3 ± 15.7	0.33

Data are given as mean, ± SD.

**Table 5 jcm-09-00653-t005:** The results of the 1-h pad test in both groups 6–12 months after TVT sling implantation, with regard to the type of reconstructive surgery.

	Control Group	Study Group	*p*	Sacrocolpopexy	Native Tissue Repair	Vaginal Mesh	Sacrocolpopexy with Native Tissue Repair	Vaginal Mesh and Native Tissue Repair	*p*
Pad test (g)	2.5 ± 22.0	1.8 ± 7.1	0.677 (ns)	0.1 ± 0.4	1.4 ± 4.4	2.1 ± 9.8	9.3 ± 18.5	1.3 ± 3.5	0.775 (ns)
Negative pad test (≤ 2 g)	234/250 (93.6%)	77/84 (91.6%)	0.574 (ns)	12/12 (100%)	29/33 (87.8%)	23/24 (95.8%)	4/5 (80%)	8/8 (100%)	ns

Data are given as mean, ± SD.

**Table 6 jcm-09-00653-t006:** The results of the IIQ-7 questionnaire in both groups 6–12 months after TVT sling implantation, with regard to the type of reconstructive surgery.

	Control Group	Study Group	*p*	Sacrocolpopexy	Native Tissue Repair	Vaginal Mesh	Sacrocolpopexy with Native Tissue Repair	Vaginal Mesh and Native Tissue Repair	*p*
IIQ-7 score	7.6 ± 16.7	8.1 ± 15.7	0.9 (ns)	2.9 ± 6.7	10 ± 6.7	6.7 ± 12.4	10.5 ± 23.3	5.2 ± 15.2	0.793 (ns)

**Table 7 jcm-09-00653-t007:** OAB symptoms before and after the sling procedure in both groups, with regard to the type of reconstructive surgery.

Parameter	Control Group	Study Group	*p*	Sacrocolpopexy	Native Tissue Repair	Vaginal Mesh	Sacrocolpopexy with Native Tissue Repair	Vaginal Mesh and Native Tissue Repair	*p*
Frequency of daytime micturition (before surgery)	7.5 ± 2.4	7.3 ± 1.7	0.86	8.0 ± 0.7	7.4 ± 1.8	7.3 ± 1.9	5.3 ± 0.6	7.3 ± 1.7	0.26
Frequency of daytime micturition (after surgery)	6.2 ± 1.7	6.0 ± 1.5	0.563	5.6 ± 1.8	6.4 ± 1.8	6.0 ± 1.2	4.0 ± 1.4	5.7 ± 0.6	0.497
*p*	<0.001	<0.001	-	0.24	0.13	0.14	ns	0.47	-
Frequency of nighttime micturition (before surgery)	1.2 ± 1.4	1.5 ± 1.2	0.07	1.0 ± 0.7	1.5 ± 1.4	1.6 ± 1.2	1.0 ± 0	0.3 ± 0.6	0.07
Frequency of nighttime micturition (after surgery)	0.8 ± 1.0	1.2 ± 1.0	0.008	0.6 ± 1.3	0.9 ± 0.7	1.8 ± 1.1	0 ± 0.0	0.7 ± 0.6	0.017
*p*	0.007	0.016	-	0.24	0.22	0.75	ns	ns	-
Episodes of urgency (before surgery)	0.5 ± 1.4	0.6 ± 1.2	0.24	0 ± 0.0	0.8 ± 1.6	0.7 ± 1.0	0 ± 0.0	0.0	0.34
Episodes of urgency (after surgery)	0.5 ± 1.5	0.2 ± 0.6	0.414	0.6 ± 1.3	0.2 ± 0.5	0.1 ± 0.3	0.0 ± 0.0	0.0 ± 0.0	0.69
*p*	0.045	0.061	-	ns	0.37	0.04	ns	ns	-

Data are given as mean, ± SD.

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
