# Peer review of "Does Previous Pelvic Organ Prolapse Surgery Influence the Effectiveness of the Sub-Urethral Sling Procedure"

_jcm, 2020, doi:10.3390/jcm9030653_

Round 1

Reviewer 1 Report

The manuscript has improved a lot and can be considered for publication. 

Reviewer 2 Report

now the manuscript can be accepted 

the revisions carried out correctly

This manuscript is a resubmission of an earlier submission. The following is a list of the peer review reports and author responses from that submission.

Round 1

Reviewer 1 Report

Thanks for the invitation to review this article, it is an interesting topic because there is little data in the literature on the right timing of sling insertion in patients with prolapse and incontinence.

Abstract

well written and comprehensive.

Introduction

well written and comprehensive.

Materials and methods

well structured

I'd like to know:

1) after how long after the first intervention, the patients of the study group underwent TVT

2) it would have been interesting to evaluate the patients who underwent surgery for prolapse without TVT and patients who in the same surgery underwent prolapse + TVT correction

3) why do you use TVT and not TOT with the first line of incontinence surgery?

4) were the patients subjected to local hormone therapy after the surgery?

5) how many patients were in menopause?

Results

well written and comprehensive.

Disussion

well structured.

when it comes to post-operative OAB, some studies show that sling can decrease OAB in patients with mixed incontinence ... insert this part

Schiavi MC, D'Oria O, Aleksa N, Vena F, Prata G, Di Tucci C, Savone D, Sciuga V, Giannini A, Meggiorini ML, Monti M, Zullo MA, Muzii L, Benedetti Panici P. Usefulness of Ospemifene in the treatment of urgency in menopausal patients affected by mixed urinary incontinence underwent mid-urethral slings surgery. Gynecol Endocrinol. 2019 Feb;35(2):155-159. doi: 10.1080/09513590.2018.1500534.

we sometimes perform the combined surgery, using native tissue repair surgery to resolve pelvic floor defects and TOT for incontinence

Is there a difference in terms of complications between patients who perform prosthetic or fascial surgery associated with sling?

Schiavi MC, Savone D, Di Mascio D, Di Tucci C, Perniola G, Zullo MA, Muzii L, Benedetti Panici P. Long-term experience of vaginal vault prolapse prevention at hysterectomy time by modified McCall culdoplasty or Shull suspension: Clinical, sexual and quality of life assessment after surgical intervention. Eur J Obstet Gynecol Reprod Biol. 2018 Apr;223:113-118. doi: 10.1016/j.ejogrb.2018.02.025.

Schiavi MC, DʼOria O, Faiano P, Prata G, Di Pinto A, Sciuga V, Colagiovanni V, Giannini A, Zullo MA, Monti M, Muzii L, Benedetti Panici P. Vaginal Native Tissue Repair for Posterior Compartment Prolapse: Long-Term Analysis of Sexual Function and Quality of Life in 151 Patients. Female Pelvic Med Reconstr Surg. 2018 Nov/Dec;24(6):419-423. doi: 10.1097/SPV.0000000000000463.

thanks and best regards

Reviewer 2 Report

The objective of the study is important, but according to the data presented no more answers can be given. The study is comparing likely different groups, those with both POP (already operated) + UI, and those with only UI. It is stated that none of the women had POP over stage 2, but information whether those with symptomatic POP were excluded is not stated. The POP status of both groups need to be shown, if there is no difference between the groups that would add the value of the study.

The language should be revised.

Abstract: Indicate first study group and then control, presentation is unprecise. Do that throughout the manuscript.

You should state clearly what is the main outcome and which ones are the secondary outcomes.

Introduction: You should add the reason why UI and POP coexist.
You report carefully the risk for de novo SUI, but you should also mention that it is possible that correction of POP may also relieve SUI (in 27% of women, Borstad et al). In the discussion you show numeral references of the topic, state shortly the existing data and then indicate why you wanted to do this study.

You need to write out LUTS when mentioned first time.

You need to add the risk of long-term pain and erosion to risk for serious adverse events.

M&M:

Indicate your study design, prospective cohort study?

Present first the study group.

You should present the baseline situation according to POP, any difference between the groups?
How long time there was from POP surgery to TVT? How many of the women had had SUI before the POP operation? How did you choose those 84 women? How many women did have POP operation in your clinic during your study time, are these the women who contacted you to have SUI after POP operation?

The follow-up time is too wide, you can`t combine 3 and 20 months, you need to take the time point where you compare the results, e.g. 12 months.

How did you get the numbers for the groups, no power calculations?

You could have used also some other questionnaire e.g. UDI-6 to get more information.

Results: Present the questions, not only numbers, if you are not familiar with the questionnaire those numbers do not say anything.

Tables: do not combine IQ7 and pad test. What do those numbers one to five present in the Tables?

Discussion: You should compare your results more actively to previous data. Discussion is too long, comprise it.